# MIRA: QUANTIFYING NEURAL NETWORK MONITORABILITY VIA FEATURE SPACE ANALYSIS

## ABSTRACT

Monitoring neural networks is increasingly important for detecting potential failures in safety-critical applications. Although out-of-distribution (OoD) detection and uncertainty estimation have been widely studied, they often rely on the assumption that neural networks learn high-quality features. However, this assumption may not hold in practice, potentially leading to undetected failures. In this work, we introduce the concept of *monitorability*, that is, the intrinsic ability of a model to highlight potential inference errors through internal activations. We provide a formal definition of monitorability and propose the *Monitorability via Input peRturbAtion (MIRA) Score*, a practical measure that quantifies this property without requiring external OoD data. Our method accounts for the behavior of the model near the decision boundary by applying norm-bounded input perturbations, and evaluates how distinguishable the resulting internal representations are by using Mahalanobis distance. Since no established baseline exists for monitorability, we validate MIRA by comparing it against the best achievable OoD detection performance across three representative methods. Through experiments across multiple architectures and domain applications, we show that the MIRA Score correlates with the strongest actual detection performance, providing a reliable tool for evaluating and comparing monitorability across different models. To the best of our knowledge, this is the first formalization and quantitative measure of monitorability. This work offers both theoretical grounding and empirical insight into the conditions under which model failures become detectable.

## 1 INTRODUCTION

Deep neural networks (DNNs) are becoming increasingly integrated into safety-critical cyber-physical systems. Applications include autonomous vehicles, medical diagnosis, and industrial automation. Ensuring the reliability and trustworthiness of these systems is therefore paramount. DNNs are typically trained on a finite dataset intended to represent the scenarios they will encounter during deployment. However, real-world scenarios are often too diverse to be fully captured by any finite training dataset (Chuang et al., 2020). In addition to this limitation, DNNs are vulnerable to adversarial attacks, where inputs can be subtly manipulated to induce incorrect predictions (Goodfellow et al., 2014). These perturbations are often imperceptible to humans but can lead the model to produce high-confidence errors. If such inputs go undetected, the consequences can be severe, particularly in applications where safety and security are crucial.

To mitigate such risks, a large body of work has focused on detecting *out-of-distribution* (OoD) inputs, which are instances that deviate from the training distribution (Hendrycks & Gimpel, 2017; Liang et al., 2018). However, many of these methods implicitly assume that the underlying model has learned semantically meaningful features (Lee et al., 2018b). This assumption does not always hold, especially when the architecture is poorly suited to the task (Bartocci & Essbai, 2024). Without validating this assumption, the reliability of these methods may be compromised. While numerous techniques have been proposed to detect OoD inputs at runtime, the literature still lacks a formal definition and metric to characterize how inherently *monitorable* a neural network is.

In this paper, we introduce the novel notion of *monitorability* for DNNs. This property captures the extent to which a model's internal representations enable the detection of anomalous or erroneous behavior at runtime. Rather than focusing on detecting anomalies, we aim to characterize how

*detectable* such anomalies are from the internal features, a property that reflects the intrinsic structure of the network. To this end, we propose the *Monitorability via Input peRturbAtion (MIRA) Score*, a practical metric that quantifies monitorability by probing the model with norm-bounded input perturbations and evaluating the separability of the resulting internal activations. To the best of our knowledge, this is the first work to formally define and quantify monitorability as a distinct property of neural networks.

**Our contribution.** We make the following contributions: 1. We formally define the concept of *monitorability* for DNNs and clarify how it differs from inference performance; 2. We propose a practical metric, the *Monitorability via Input peRturbAtion (MIRA) Score*, which quantifies monitorability; 3. We conduct experiments across multiple architectures, datasets, and data modalities, and show that the MIRA Score correlates with monitoring performance.

**Paper organization.** In Section 2, we introduce the necessary background. In Section 3, we motivate and define the notion of *monitorability* and present our MIRA score. Section 4 provides the experimental results. We discuss related work in Section 5, and conclude in Section 6.

## 2 PRELIMINARIES

This section introduces the terminology and notation used throughout the paper.

**Neural Networks.** In this work, we focus on neural networks (NNs) applied to classification tasks, although our concepts generalize directly to other settings. We denote by $f : \mathcal{X} \subseteq \mathbb{R}^d \to \mathcal{Y} \subseteq \mathbb{R}^C$ a NN mapping a $d$-dimensional input $x \in \mathcal{X}$ to a vector of logits over $C$ classes, with predicted class $class(f(x)) = \arg\max_{i \in \mathcal{C}} f(x)$, where $\mathcal{C} = 0, 1, \ldots, C-1$ is the label set. We further denote by $f^l : \mathbb{R}^d \to \mathbb{R}^{n_l}$ the function computing the output at the $l$-th layer of the network, where $n_l$ denotes the number of neurons in layer $l$.

**Out-of-Distribution Detection.** In a standard supervised learning setting, a model is trained on a dataset $\mathcal{D}_{in}^{train} = \{(x_i, y_i)\}_{i=1}^N$ sampled from an unknown joint distribution $\mathcal{P}_{in}(x, y)$ over $\mathcal{X} \times \mathcal{Y}$. We denote by $\mathcal{P}_{in}$ the probability distribution of the *in-distribution* (ID) data, and by $\mathcal{P}_{in}(x)$ its marginal distribution over inputs. Out-of-distribution (OoD) detection aims to identify test-time inputs that do not resemble the training distribution, i.e., inputs $x$ that are not from $\mathcal{P}_{in}(x)$. This is particularly important in safety-critical applications, where anomalous inputs can lead to severe failures. However, it is important to note that misclassifications may also occur for ID inputs, which is a distinct scenario not directly addressed by OoD detection.

**Mahalanobis distance.** Mahalanobis distance measures the distance between a point and a probability distribution (Mahalanobis, 1936). Formally, the distance between a point $x$ and a probability distribution with mean $\mu$ and covariance matrix $\Sigma$ is defined as:

$$D_M(x) = \sqrt{(x - \mu)^T \Sigma^{-1} (x - \mu)}. \tag{1}$$

Unlike Euclidean distance, Mahalanobis distance accounts for correlations among variables and scales according to the distribution's variance, making it particularly useful for detecting outliers in multivariate settings. In the context of deep learning, Lee et al. (2018b) demonstrated that Mahalanobis distance can be effectively used for OoD detection. They further showed that features extracted by trained NNs support the assumption of Gaussian Discriminant Analysis (GDA). Under this assumption, the softmax classifier becomes equivalent to selecting the class whose mean is closest in Mahalanobis distance. Therefore, the Mahalanobis score aligns well with the underlying probabilistic structure of the classifier.

## 3 MONITORABILITY

In this study, we propose *monitorability* as an intrinsic property of neural networks. Our goal is to give a definition for this property and provide a quantitative measure of it.

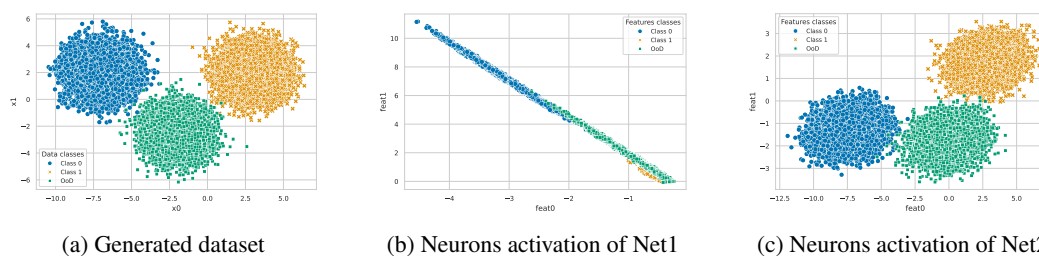

| (a) Generated dataset | (b) Neurons activation of Net1 | (c) Neurons activation of Net2 |

Figure 1: Illustration of monitorability using a toy dataset. (a) A synthetic two-dimensional dataset with two ID classes and one OoD class. (b–c) Penultimate-layer activations of two different NNs trained on the same data. Although both models achieve $100\%$ accuracy on ID data, they differ in how they represent OoD samples in feature space. Net2 (c) learns a more separable representation for OoD data than Net1 (b). Therefore, we consider Net2 more monitorable.

## 3.1 MOTIVATION

Out-of-distribution (OoD) detection aims to flag inputs unlikely under $\mathcal{P}_{in}(x)$, typically using either logits or hidden representations (Liang et al., 2018; Liu et al., 2020; Sun et al., 2022; Lee et al., 2018b; Hashemi et al., 2023; Henzinger et al., 2020b). While often effective, these methods implicitly assume that learned features reliably separate ID from OoD inputs. In practice, neural networks can still yield high-confidence predictions on OoD or adversarial inputs (Nguyen et al., 2015; Goodfellow et al., 2014), where errors arise from local boundary irregularities rather than distributional shifts, limiting conventional detectors.

Existing approaches detect *when* a model may fail, but not whether such failures are *detectable at runtime* using the model's internal representation. In this work, we introduce and formalize the notion of *monitorability* as the intrinsic ability of a model to be monitored using accessible features, even when misclassifications occur on ID inputs.

To better illustrate our intuition, we train two NNs on the toy dataset shown in Figure 1a. The dataset contains two ID classes and a third class, considered OoD. The two models Net1 and Net2 are both trained on the ID data only, achieving $100\%$ classification accuracy (architectures available in Appendix A). We then evaluate them on the OoD samples and visualize the activation patterns at the penultimate layer, shown in Figures 1b and 1c for Net1 and Net2, respectively. Despite identical classification performance on the ID data, the two networks exhibit significantly different feature representations, particularly in how they separate OoD inputs. This suggests that even similarly accurate models can differ substantially in terms of monitorability.

## 3.2 FORMAL DEFINITION OF MONITORABILITY

We propose *monitorability* as an intrinsic property of neural networks, formalizing the extent to which their failures can be detected at runtime.

**Definition 1** (Monitorability). Let $\mathcal{P}_{in}(x, y)$ be a data distribution over $\mathcal{X} \times \mathcal{Y}$, where $\mathcal{X} \subseteq \mathbb{R}^d$ and $\mathcal{Y} \subseteq \mathbb{R}^m$.

Let $f : \mathcal{X} \to \mathcal{Y}$ a neural network and $\mathcal{L} : \mathcal{Y} \times \mathcal{Y} \to \mathbb{R}_{\geq 0}$ a task-specific loss function. Then $f$ is said to be *l-monitorable* if there exists a set $Z^l \subseteq \mathbb{R}^{n_l}$ and a finite $\epsilon \geq 0$ such that:

$$\forall (x, y) \sim \mathcal{P}_{in}, \quad \mathcal{L}(f(x), y) \leq \epsilon \iff f^l(x) \in Z^l,$$

where $f^l(x)$ is the output at the $l$-th layer. The threshold $\epsilon$ is a task-specific constant and it must be chosen such that $\mathcal{L}(f(x), y) \leq \epsilon$ implies a correct prediction: e.g., for cross-entropy loss, this requires $\epsilon < \log(C)$, where $C$ is the number of classes.

Intuitively, a model is monitorable if prediction quality can be inferred from some internal layer. It is worth noting that $Z^l$ may be arbitrarily complex. Under this formulation, the monitoring problem corresponds to identifying such a layer and set. For instance, with a Mahalanobis-based monitor (Lee et al., 2018b), $Z^l = \{p \in \mathbb{R}^{n_l} : D_M(p) < \tau^l\}$. The employment of a general loss function

$\mathcal{L}$ allows the definition to apply regardless of the output type. In classification, $\mathcal{L}$ could be the cross-entropy loss, while in regression it could be the mean squared error.

### 3.3 PROPOSED METRIC

Definition 1 provides an abstract formalization of monitorability, but it does not quantify how monitorable a neural network is. To address this, we propose a metric that estimates this property.

Our key idea is to evaluate how well erroneous predictions can be distinguished based on internal activations at a given layer $l$. Intuitively, as illustrated in Figure 1, we aim to quantify whether we are in a scenario resembling Figure 1b (low monitorability) or Figure 1c (high monitorability). While labeled OoD data make this comparison straightforward, we seek a metric linked to the model's own behavior and not to external datasets.

To this end, we introduce controlled perturbations of ID samples. Given a dataset $D$, a radius $\epsilon > 0$, and $p \in [1, \infty]$, we define an $\ell_p$–perturbed dataset as

$$\tilde{D} = \left\{ x + \delta(x, \epsilon) : x \in D, \ \|\delta(x, \epsilon)\|_p \leq \epsilon \right\}. \tag{2}$$

The perturbation $\delta(x, \epsilon)$ can be arbitrary as long as it moves $x$ toward the decision boundary, potentially crossing it. Our approach aligns with the findings of Lee et al. (2018a), which suggest that local boundary behavior can generalize to unseen shifts. We then evaluate monitorability by assessing how separable the perturbed features $f^l(x + \delta)$ are from the unperturbed ones.

To measure separability we use the Mahalanobis distance $D_M$ (1). However, $D_M$ is not directly comparable across layers with different dimensionalities: under the Gaussian assumption, $D_M$ follows a $\chi^2$ distribution with $d = \dim(f^l(x))$ degrees of freedom, whose expectation grows with $d$. To obtain a dimension-calibrated and unbounded scale, we convert $D_M$ into a *surprisal score*:

$$S(f^l(\tilde{x})) \ = \ -\log\left(\mathrm{sf}_{\chi_d^2}(D_M(f^l(\tilde{x})))\right), \tag{3}$$

where $\mathrm{sf}_{\chi_d^2}$ is the chi-square survival function. Intuitively, $S$ quantifies how surprising a feature vector is under the assumed class distribution. We are now ready to define our score.

**Definition 2** (MIRA Score). The *Monitorability via Input peRturbAtion (MIRA)* score for a classifier $f$ on dataset $D$ at layer $l$ is:

$$MIRA(f, D, l) = \frac{1}{S_0} \int_{\epsilon_{\min}}^{\epsilon_{\max}} \mathbb{E}_{\tilde{x} \sim \tilde{D}}\big[S(f^l(\tilde{x})) - S_0\big] \ p(\epsilon) \, d\epsilon, \tag{4}$$

where $S(\cdot)$ is the surprisal score defined above, $S_0 = \mathbb{E}_{x \sim D}\big[S(f^l(x))\big]$, $\mathcal{C}$ is the set of classes on which the classifier $f$ is trained, and $p(\epsilon)$ is a user-defined probability distribution over perturbation magnitudes that reflects their relative importance. In particular, $p(\epsilon) \geq 0$ and $\int_{\epsilon_{\min}}^{\epsilon_{\max}} p(\epsilon) \, d\epsilon = 1$. For example, choosing $p(\epsilon)$ uniform assigns equal importance to all perturbation magnitudes.

This metric captures the extent to which a model's internal features allow for error detection in regions of high predictive risk by integrating over a range of perturbation magnitudes. This integration allows us to characterize the change in the model's behavior across different regions of the input space. It is conceptually simple and requires no external OoD data. Importantly, MIRA is not designed as a runtime detection method but as a pre-deployment evaluation metric that can be computed efficiently using only ID data.

## 4 EXPERIMENTAL EVALUATION

We evaluate the proposed MIRA score across three data modalities: computer vision, tabular data, and natural language processing (NLP). In all cases, we compute MIRA on the penultimate layer of each model, as this layer typically encodes the most informative features prior to classification.

### 4.1 EVALUATION PROTOCOL

The lack of a baseline for measuring monitorability is a key challenge to evaluation. Monitorability is an intrinsic property of the model, whereas monitoring performance depends on the choice and

calibration of a specific monitoring method. To bridge this gap, we use OoD detection performance as a *proxy* for monitorability. Specifically, for each dataset we report the best achievable detection performance across three representative methods: ODIN (Liang et al., 2018), Mahalanobis distance (Lee et al., 2018b), and Energy-based scoring (Liu et al., 2020). These methods are grounded in fundamentally different principles, namely confidence calibration, distance in feature space, and energy modeling, which together provide a diverse basis for evaluation. Fine-tuning procedures and implementation details are provided in Appendix B.5.

This "best-of" aggregation approximates the most favorable monitoring potential a model can offer, which is precisely what monitorability is meant to capture: whether the model's representational structure enables effective monitoring *in principle*, independent of detector-specific limitations.

Our experiments are guided by the following research questions:

**RQ1:** Does MIRA Score correlate with the best achievable OoD detection performance across diverse models, thereby validating it as a faithful proxy for monitorability?

**RQ2:** Does this relationship hold consistently across different data modalities?

**RQ3:** Can MIRA provide insights into a model's monitoring potential that are not tied to the limitations of any single detection method?

**RQ4:** Can MIRA be computed efficiently at pre-deployment time so that practitioners can use it to guide model selection without access to OoD data or detector-specific tuning?

### 4.2 PERTURBATION SETUP

A crucial design decision is the perturbation used to generate $D_\epsilon$. In this context, the strength of the attack is not critical: perturbations do not need to be "subtle" (e.g., imperceptible to the human eye, in the case of images), but they should rather provide a meaningful direction toward the boundary. Although strong adversarial methods such as the Carlini-Wagner attack (Carlini & Wagner, 2017) could be used, their computational cost is unnecessarily high for our purposes. We consider the Fast Gradient Sign Method (FGSM) (Goodfellow et al., 2014) a better choice, since it efficiently produces perturbations in the required direction and allows us to probe local sensitivity at scale.

To make these perturbations comparable across models, we define the perturbation interval $[\epsilon_{\min}, \epsilon_{\max}]$ using a consistent strategy. Here, $\epsilon_{\min}$ is chosen as the smallest value that reduces accuracy to a certain threshold, and $\epsilon_{\max} = 2 \cdot \epsilon_{\min}$. It is important to use the same threshold across the compared models. The detailed choice procedure is described in Appendix B.6.

### 4.3 EXPERIMENTAL SETUP

#### 4.3.1 DATASETS

**Computer Vision.** CIFAR-10 and CIFAR-100 (Krizhevsky et al., 2009) benchmarks are used as ID datasets. OoD evaluation is conducted on iSUN (Xu et al., 2015), GTSRB (Stallkamp et al., 2011), SVHN (Netzer et al., 2011), TinyImageNet (Le & Yang, 2015), Places365 (Zhou et al., 2018), Textures (Cimpoi et al., 2014), and Gaussian-noised CIFAR. The iSUN and TinyImageNet datasets were obtained from the ODIN repository.[1]

**Tabular.** We use the Sensorless Drive Diagnosis dataset (Bator, 2013). Models are trained on the first five classes, with the remaining classes (5–10) used for OoD evaluation.

**NLP.** We finetune on SST-2 (Socher et al., 2013) and evaluate OoD performance on IMDB (Maas et al., 2011), News Category (Misra, 2022), Yelp Polarity Review (Zhang et al., 2015), and Twitter Topic Classification (Antypas et al., 2022).

#### 4.3.2 MODELS

**Computer Vision.** We employ ResNet-18 (He et al., 2016), DenseNet (Huang et al., 2017a), a lightweight custom CNN, and a finetuned Vision Transformer (ViT) (Dosovitskiy et al., 2020) re-

---

[1] https://github.com/facebookresearch/odin/tree/main

Table 1: OoD detection performance (AUROC %) using ODIN, Mahalanobis distance, and Energy-based scoring. Models are trained on CIFAR-10 and CIFAR-100 and evaluated on multiple OoD datasets (columns). The last column reports the average of the AUROC scores among the three monitoring methods. The last row in each block shows the MIRA Score computed with respect to the penultimate layer. Higher MIRA Scores consistently align with better global detection performance, highlighting a correlation between monitorability and OoD detection capability.

| | | | GSTRB | SVHN | iSUN | TinyImageNet | Textures | Places365 | Gaussian | Average |
|---|---|---|---|---|---|---|---|---|---|---|
| **CIFAR-10** | ResNet-18 | ODIN | 81.97 | 83.48 | 92.21 | 89.0 | 76.47 | 76.91 | 73.43 | |
| | | Mahalanobis | **94.79** | **95.81** | **95.08** | **94.78** | **94.71** | 83.73 | **98.39** | **94.78** |
| | | Energy | 93.25 | 93.46 | 94.98 | 93.34 | 89.94 | **89.91** | 88.71 | |
| | | MIRA Score | | | | 6.0549 | | | | |
| | DenseNet | ODIN | 91.09 | 91.26 | **99.15** | **98.84** | 82.83 | 90.62 | 98.29 | |
| | | Mahalanobis | 89.54 | **96.54** | 94.2 | 93.85 | **93.05** | 64.49 | **99.21** | **96.14** |
| | | Energy | **94.41** | 92.66 | 98.59 | 98.1 | 86.8 | **91.81** | 97.63 | |
| | | MIRA Score | | | | 16.0107 | | | | |
| | CustomNet | ODIN | **79.78** | **89.21** | 60.88 | 62.19 | **82.13** | **82.95** | **78.39** | |
| | | Mahalanobis | 44.11 | 16.41 | **78.20** | **76.79** | 29.35 | 39.39 | 44.99 | **81.06** |
| | | Energy | 78.96 | 85.05 | 68.73 | 68.38 | 78.73 | 81.31 | 77.91 | |
| | | MIRA Score | | | | $-0.07176$ | | | | |
| | ViT | ODIN | 99.71 | 99.77 | 96.44 | 96.29 | 99.98 | 99.18 | 98.87 | |
| | | Mahalanobis | **99.92** | **99.97** | **99.80** | **99.45** | **100.0** | **99.92** | **99.96** | **99.86** |
| | | Energy | 99.91 | 99.89 | 98.47 | 97.6 | 99.97 | 98.67 | 98.06 | |
| | | MIRA Score | | | | 89.2544 | | | | |
| **CIFAR-100** | ResNet-18 | ODIN | 90.20 | 88.17 | **94.59** | **94.74** | **87.07** | 83.79 | 91.67 | |
| | | Mahalanobis | 71.97 | 85.23 | 86.68 | 87.12 | 85.39 | 71.49 | **97.17** | **91.27** |
| | | Energy | **91.35** | **89.80** | 90.24 | 90.16 | 86.32 | **84.19** | 83.86 | |
| | | MIRA Score | | | | 0.6572 | | | | |
| | DenseNet | ODIN | **94.58** | **92.85** | 87.4 | 88.48 | 81.00 | **84.82** | **99.65** | |
| | | Mahalanobis | 77.19 | 86.76 | **95.41** | **95.41** | **99.77** | 54.65 | 97.51 | **94.64** |
| | | Energy | 92.84 | 92.38 | 81.52 | 82.48 | 79.06 | 83.72 | 97.51 | |
| | | MIRA Score | | | | 2.8060 | | | | |
| | ViT | ODIN | 91.56 | 96.20 | 93.49 | 95.21 | 99.89 | 99.10 | 99.46 | |
| | | Mahalanobis | **95.25** | **99.11** | **98.10** | **98.32** | **99.97** | **99.74** | **99.93** | **98.63** |
| | | Energy | 92.48 | 97.18 | 91.77 | 94.00 | 99.49 | 96.90 | 97.56 | |
| | | MIRA Score | | | | 53.2344 | | | | |

trieved from Hugging Face (Face & Research, 2023), pretrained on ImageNet-21k (Deng et al., 2009).. Pretrained weights are taken from OpenOOD (Zhang et al., 2023)[2] and ODIN (Liang et al., 2017).[1] Additional architectural details are reported in Appendix B.2.

**Tabular.** We use three MLPs with different depths/widths and two transformer-based models (1-layer and 2-layer encoders). Architectures and training details are available in Appendix B.3.

**NLP.** We finetune different pretrained transformer-based models, with varying size and performance. In particular, we use RoBERTa (Liu et al., 2019), DistilBERT (Sanh et al., 2019), ELECTRA (Clark et al., 2020), and DeBERTaV3 (He et al., 2021). Details are reported in Appendix B.4.

## 4.4 RESULTS AND DISCUSSION

**Computer Vision.** Results are reported in Table 1. Higher MIRA scores consistently align with stronger OoD detection: e.g., ViT achieves both the highest MIRA (89.25) and the strongest AUROC (~99%), while CustomNet has the lowest MIRA ($-0.07$) and weakest monitoring. Notably, the MIRA can assume negative values, and this would mean that the perturbed data are less detectable that ID data, showcasing very bad monitoring capabilities.

---

[2]https://github.com/Jingkang50/OpenOOD

Table 2: OoD detection performance (AUROC %) using ODIN, Mahalanobis distance, and Energy-based scoring. The evaluation involves using classes 6-11 from the Sensorless Drive Diagnosis as OoD. The table structure is the same as Table 1.

| | | Class 6 | Class 7 | Class 8 | Class 9 | Class 10 | Class 11 | Average |
|---|---|---|---|---|---|---|---|---|
| MLP | ODIN | 72.06 | 0.08 | 54.92 | 72.23 | 28.76 | 0.0 | |
| | Mahalanobis | **85.35** | **99.56** | **91.13** | **95.17** | **73.58** | **100**.0 | **90.80** |
| | Energy | 76.44 | 0.12 | 51.35 | 78.62 | 29.32 | 0.0 | |
| | MIRA Score | | | | 30.1594 | | | |
| DeepMLP | ODIN | 60.32 | 0.35 | 25.36 | 61.32 | 54.42 | 0.01 | |
| | Mahalanobis | **84.41** | **97.21** | **89.03** | **85.82** | **61.00** | **99.98** | **86.24** |
| | Energy | 70.14 | 0.44 | 25.88 | 73.73 | 35.12 | 0.01 | |
| | MIRA Score | | | | 12.7098 | | | |
| WideMLP | ODIN | 66.31 | 0.04 | 30.41 | 70.66 | 42.44 | 0.00 | |
| | Mahalanobis | **89.94** | **99.81** | **90.99** | **97.65** | **78.91** | **99.98** | **92.88** |
| | Energy | 70.24 | 0.18 | 30.95 | 74.48 | 35.76 | 0.00 | |
| | MIRA Score | | | | 63.5103 | | | |
| Transformer | ODIN | 15.73 | 62.64 | 68.06 | 25.54 | **80.14** | **91.46** | |
| | Mahalanobis | **78.71** | 76.88 | 79.88 | **88.61** | 44.48 | 79.71 | **85.36** |
| | Energy | 68.07 | **90.02** | **83.27** | 71.4 | 42.72 | 56.93 | |
| | MIRA Score | | | | 7.8706 | | | |
| DeepTransformer | ODIN | 54.81 | 41.82 | **88.59** | 47.42 | 14.93 | 28.18 | |
| | Mahalanobis | **69.81** | **75.71** | 56.88 | **79.89** | 47.02 | **87.50** | **77.86** |
| | Energy | 20.64 | 66.41 | 35.99 | 20.25 | **65.67** | 47.02 | |
| | MIRA Score | | | | 4.3676 | | | |

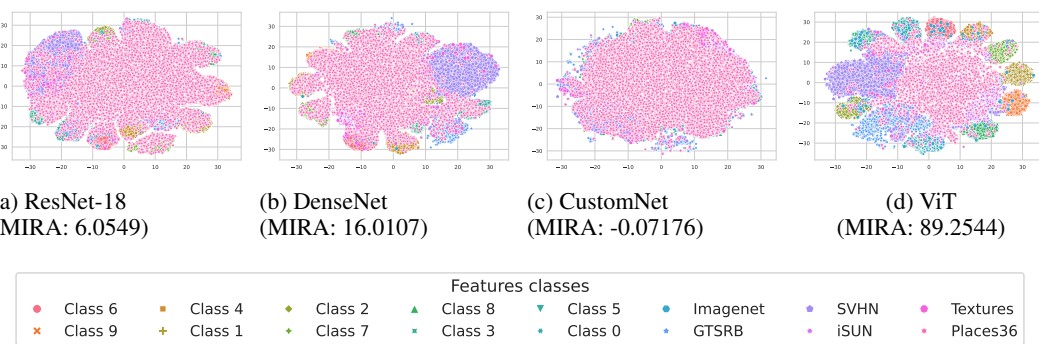

(a) ResNet-18
(MIRA: 6.0549)

(b) DenseNet
(MIRA: 16.0107)

(c) CustomNet
(MIRA: -0.07176)

(d) ViT
(MIRA: 89.2544)

Features classes

| • Class 6 | ▪ Class 4 | ♦ Class 2 | ▲ Class 8 | ▼ Class 5 | • Imagenet | • SVHN | • Textures |
| × Class 9 | + Class 1 | ✦ Class 7 | ⨯ Class 3 | ✳ Class 0 | • GTSRB | • iSUN | • Places36 |

Figure 2: t-SNE visualization of penultimate-layer activations on CIFAR-10 for ResNet-18, DenseNet, CustomNet, and ViT. Points are colored by class label. Better cluster separation aligns with higher MIRA, showing that the score is linked to the structure of the learned feature space.

**Tabular Data.** Results are shown in Table 2. MIRA successfully distinguishes models with varying monitoring potential: WideMLP achieves a high MIRA (63.5) and correspondingly strong OoD detection, while the DeepTransformer yields both low MIRA (4.37) and weaker performance.

**NLP.** Table 3 reports MIRA scores for the NLP models. DeBERTaV3 attains the highest monitorability score, consistent with the intuition that recent and more expressive architectures provide richer feature representations that facilitate monitoring. At the other end, DistilBERT yields the weakest score, in line with its reduced expressiveness and size. Importantly, these trends are reflected in the average OoD detection performance, which follows the same ordering.

Table 3: OoD detection performance (AUROC %) using ODIN, Mahalanobis distance, and Energy-based scoring. The evaluation involves different OoD datasets for models finetuned on SST-2. The table structure is the same as Table 1.

| | | IMDB | News Category | Yelp | Twitter | Average |
|---|---|---|---|---|---|---|
| RoBERTa | ODIN | 65.77 | 79.40 | 59.67 | 73.49 | |
| | Mahalanobis | **70.98** | **86.47** | **70.06** | **81.13** | **77.16** |
| | Energy | 66.65 | 80.56 | 60.23 | 75.26 | |
| | MIRA Score | | | 2632.9362 | | |
| DistilBERT | ODIN | 64.35 | 78.09 | 65.01 | 79.44 | |
| | Mahalanobis | **65.98** | **83.24** | **70.80** | **86.12** | **76.54** |
| | Energy | 64.86 | 78.43 | 66.22 | 80.13 | |
| | MIRA Score | | | 2015.6614 | | |
| ELECTRA | ODIN | 65.11 | 77.04 | 58.05 | 76.97 | |
| | Mahalanobis | **76.44** | **86.27** | **71.13** | **88.62** | **80.61** |
| | Energy | 64.71 | 77.18 | 57.56 | 76.71 | |
| | MIRA Score | | | 3636.6820 | | |
| DeBERTaV3 | ODIN | 74.48 | 83.01 | 70.27 | 79.61 | |
| | Mahalanobis | **83.47** | **91.65** | **83.57** | **86.74** | **86.29** |
| | Energy | 74.06 | 82.68 | 69.67 | 79.19 | |
| | MIRA Score | | | 3793.6124 | | |

**Visualizing Monitorability in Feature Space.** To complement the quantitative results, we visualize the feature spaces induced by different models using t-SNE (van der Maaten & Hinton, 2008) projections of penultimate-layer activations. We extract activations for both ID and OoD samples and project them into 2D. Figure 2 shows the resulting plots for the models trained on CIFAR-10. The visualizations reveal clear differences across architectures: ResNet-18 and DenseNet produce moderately structured feature spaces with partial class separation, ViT generates highly compact and well-separated clusters, while CustomNet exhibits substantial overlap across classes.

These results provide an intuitive perspective on monitorability: higher MIRA scores correspond to more organized feature representations, whereas lower scores reflect entangled and poorly separated spaces. MIRA thus quantifies this structural property in a principled way without access to the OoD datasets. Additional visualizations for the tabular and NLP modalities are provided in Appendix B.

**Discussion.** Across all domains, MIRA exhibits good correlation with the best achievable OoD detection performance (RQ1). This trend is consistent across modalities (RQ2), with particularly clear separation between architectures of different capacity. Moreover, MIRA captures intrinsic monitoring potential even when individual detectors disagree, demonstrating its detector-agnostic nature (RQ3). For example, in the vision experiments (Table 1), the Mahalanobis detector failed with Places365 for DenseNet, yet the other methods were still achieving performance in line with the average. Finally, because MIRA requires only ID data and efficient FGSM perturbations, it can be computed pre-deployment efficiently, making it practical for guiding model selection in real-world workflows (RQ4). In fact, tuning multiple OoD detectors through a grid search and subsequent evaluation on several OoD datasets is much more expensive than computing the MIRA Score.

## 5 RELATED WORK

**Neural network verification.** A significant body of work focuses on the formal verification of neural networks, particularly with respect to the *robustness* property. Most approaches focus automated techniques to prove properties like local or global robustness (Tran et al., 2022; Seshia et al., 2018; Katz et al., 2017; Huang et al., 2017b; Gopinath et al., 2018; Athavale et al., 2024; Wang et al.,

2021b; Leino et al., 2021). Recent advances have significantly improved the scalability of these methods, enabling verification of models with thousands of neurons (Wang et al., 2021a; Zhang et al., 2018). In addition to verification-based approaches, several works also propose quantitative robustness metrics. For example, Hendrycks & Dietterich (2019); Arcaini et al. (2022) examine robustness under natural distributional shifts such as image corruptions, while Bastani et al. (2016); Levy & Katz (2022) focus on robustness against adversarial perturbations. Weng et al. (2018) introduces a robustness metric based on extreme value theory, which estimates the minimum input distortion required to alter a model's prediction.

**OoD detection.** Static verification techniques are not designed to detect anomalies at runtime. OoD detection focuses on identifying anomalous inputs during inference. Methods include confidence-based scoring (e.g., ODIN (Liang et al., 2018)), distance-based scoring in feature space (Lee et al., 2018b), and energy-based techniques (Liu et al., 2020). Others methods rely on auxiliary OoD datasets (Hendrycks et al., 2018) or using generative models (Lee et al., 2017). Although these techniques can be effective in detecting anomalous inputs, they do not evaluate how reliably the internal representations of a model can be used to detect potential failures.

**Activation pattern monitoring.** Neural activation patterns (NAPs) provide insight into how information propagates through a model and how decisions are made (Bäuerle et al., 2022; Yosinski et al., 2015; Zeiler & Fergus, 2014). Several runtime monitoring methods build on this idea by abstracting activation behavior observed during training (Cheng et al., 2023). (Cheng et al., 2019) propose a binary abstraction that accepts inputs only if their NAPs exactly match those seen during training, but this lacks scalability and generalization. A more effective strategy is the *box abstraction* (Henzinger et al., 2020b; Cheng et al., 2020), which records per-layer activation ranges and checks whether test-time activations fall within them. Hashemi et al. (2021) further improve this by modeling activations as Gaussians and using statistical confidence bounds to define valid regions.

## 6 CONCLUSION

We introduced the concept of *monitorability* for DNNs and proposed the MIRA Score, a practical metric that quantifies a model's intrinsic ability to support runtime error detection. By studying a model's behavior through norm-bounded input perturbations and analyzing the resulting feature-space, MIRA offers a principled and architecture-independent means of assessing monitorability. Experimental results across multiple models and OoD detection methods demonstrate a strong correlation between MIRA and actual detection performance. Beyond model-level evaluation, MIRA can guide design decisions such as selecting the most suitable layer for feature-based monitoring, which is a crucial aspect for such methods (Henzinger et al., 2020a; Kueffner et al., 2023; Lee et al., 2018b), and identifying class-specific vulnerabilities where monitorability may be weak. To the best of our knowledge, no prior work has formally addressed this problem or proposed such a notion.

A current limitation is our empirical procedure for selecting the perturbation range (Section 4.2). As future work, we plan to explore more principled strategies, including model-adaptive and formal methods approaches, to better align the range with decision boundary sensitivity. Moreover, we aim to extend our concept of monitorability beyond single-layer activations. We believe that modeling inter-layer dynamics could lead to a richer, more general notion of monitorability. We also aim to extend the MIRA framework to support formal verification pipelines by incorporating monitorability as a verifiable property. Additionally, we intend to investigate the use of alternative norms in the computation of MIRA, and to study how different norm choices influence both the metric and its interpretation.

**Reproducibility Statement** We used standard public datasets without additional preprocessing, ensuring accessibility and consistency. All algorithms are deterministic, and we fixed random seeds across all experiments to facilitate replication. Details on the monitorability definition and metric are provided in Section 3, while experimental setup and hyperparameters are summarized in Section 4 and Appendix B. The code used for our experiments is available online.[3]

---

[3] https://anonymous.4open.science/r/monitorability_metric-4E80

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

## A  Toy example

The first model (Net1) has three fully connected layers with two neurons each; the second one (Net2) has two fully connected layers, also with two neurons each. We train both models on the ID data only, achieving $100\%$ classification accuracy.

## B  Experimental evaluation

### B.1  Environment, implementation, and resource usage.

Experiments were conducted on an AIME workstation equipped with an AMD Ryzen Threadripper 3990X CPU, 256 GB DDR4 RAM, and four NVIDIA GeForce RTX 3090 GPUs, running Ubuntu 22.04.4 LTS. All experiments were implemented in Python 3.8 using PyTorch 2.1.0 (Paszke et al., 2017). The MIRA Score involves computing an integral, which we approximate using 30 uniformly sampled perturbation values. Despite the integration, computation remains efficient: scoring takes 2–10 minutes per model. OoD method evaluation about ranges from few minutes for tabular data models, around 1 hour for vision models and 3 hours for NLP modeles. Peak GPU memory usage is around 20 GB, primarily due to dataset loading. Lower batch sizes can reduce memory demands if needed.

### B.2  Computer Vision

**CustomNet**  Table 4 shows the architecture of our lightweight CNN, which was introduced in Cheng et al. (2019).

The model was trained for 200 epochs using RMSProp with an initial learning rate of 0.001. Training was conducted on a dataset of 50,000 CIFAR-10 samples, with evaluation on a 10,000-sample test set.

**Models**  Table 5 reports the accuracy of all models used in our vision experiments.

### B.3  Tabular Data

We evaluate feed-forward multilayer perceptrons (MLPs) as well as Transformer-based architectures. Architectures are implemented in PyTorch, with details given below. All models achieve near-perfect accuracy, making this benchmark useful for stress-testing monitorability. The models are trained for 30 epochs with learning rate 0.001.

**MLP Architectures.**

- **MLP**: 2 hidden layers 32 neurons each, ReLU activation.
- **DeepMLP**: 4 hidden layers 16 neurons each, ReLU activation.
- **WideMLP**: 1 hidden layer with 64 neurons, ReLU activation.

Table 4: Architecture of the custom CNN used for CIFAR-10.

| Layer | Description |
|---|---|
| Conv Layer 1 | Conv2D (40 filters) + BatchNorm + ReLU |
| Pooling 1 | MaxPooling2D |
| Conv Layer 2 | Conv2D (20 filters) + BatchNorm + ReLU |
| Pooling 2 | MaxPooling2D |
| FC Layer 1 | Fully connected (240 units) + ReLU |
| FC Layer 2 | Fully connected (84 units) + ReLU |
| Output Layer | Fully connected (10 units, no activation) |

Table 5: Vision models used in the evaluation and assessment of the MIRA score. The "Network" column indicates the backbone architecture.

| ID dataset | Network | Accuracy (%) | |
|---|---|---|---|
| | | *train* | *test* |
| CIFAR-10 | ResNet-18 | 100.00 | 95.17 |
| | DenseNet | 100.00 | 95.19 |
| | CustomNet | 98.85 | 69.50 |
| | ViT | 99.40 | 97.66 |
| CIFAR-100 | ResNet-18 | 99.97 | 77.27 |
| | DenseNet | 97.00 | 77.64 |
| | ViT | 98.39 | 90.38 |

Table 6: Tabular models evaluated on the Sensorless Drive Diagnosis dataset.

| Network | Accuracy (%) | |
|---|---|---|
| | *train* | *test* |
| MLP | 99.99 | 99.82 |
| DeepMLP | 99.81 | 99.54 |
| WideMLP | 99.99 | 99.80 |
| Transformer | 99.99 | 99.94 |
| DeepTransformer | 99.89 | 99.84 |

**Transformer Architectures.** We also implement a Transformer encoder, which tokenizes each input feature into a $d_{\text{model}}$-dimensional embedding with learnable per-feature weights. A learnable [CLS] token is prepended, followed by Transformer encoder layers with GELU activation and pre-norm. The [CLS] embedding is passed through a linear classifier head.

Two variants are trained:

- **Transformer**: $d_{\text{model}} = 8$, $n_{\text{head}} = 2$, 2 layers, feedforward dimension 64.

- **DeepTransformer**: same configuration but with an additional transformer layer.

**Accuracies.** Table 6 reports the train/test accuracies.

B.4 NATURAL LANGUAGE PROCESSING

For natural language understanding, we finetune four pretrained Transformer models: RoBERTa$_{\text{base}}$, DistilBERT, ELECTRA, and DeBERTaV3. Training is conducted on the GLUE SST-2 dataset using HuggingFace's Transformers library. All models use AdamW optimizer and linear learning rate decay and 500 warmup steps and 3 epochs. The learning rates are:

- **RoBERTa**: learning rate $2 \times 10^{-5}$.

- **DistilBERT**: learning rate $3 \times 10^{-5}$.

- **ELECTRA**: learning rate $1 \times 10^{-5}$.

- **DeBERTaV3**: learning rate $2 \times 10^{-5}$.

Table 7 summarizes performance on SST-2.

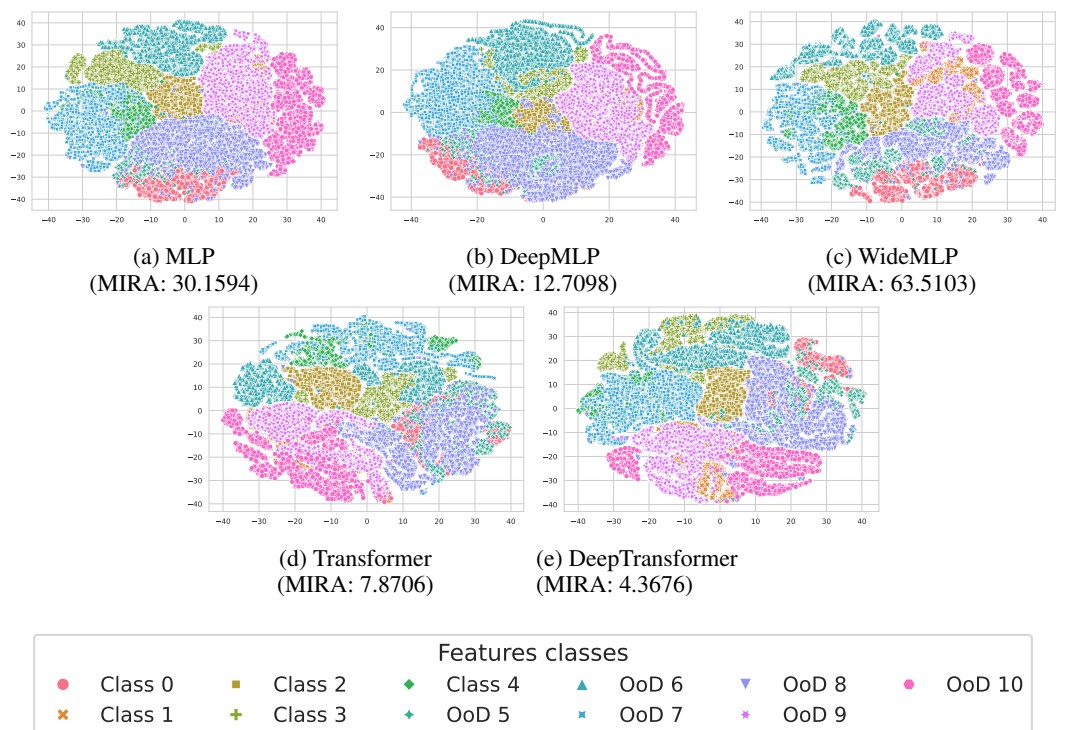

(a) MLP
(MIRA: 30.1594)

(b) DeepMLP
(MIRA: 12.7098)

(c) WideMLP
(MIRA: 63.5103)

(d) Transformer
(MIRA: 7.8706)

(e) DeepTransformer
(MIRA: 4.3676)

Features classes

| | | | |
|---|---|---|---|
| ● Class 0 | ■ Class 2 | ◆ Class 4 | ▲ OoD 6 | ▼ OoD 8 | ● OoD 10 |
| ✕ Class 1 | + Class 3 | ✦ OoD 5 | ✕ OoD 7 | ✳ OoD 9 | |

Figure 3: t-SNE visualization of penultimate-layer activations on Sensorless Drive Diagnosis. Better cluster separation aligns with higher MIRA, showing that the score is linked to the structure of the learned feature space.

Table 7: Language models finetuned on SST-2.

| Network | Accuracy (%) | |
|---|---|---|
| | *train* | *test* |
| RoBERTa | 98.35 | 93.35 |
| DistilBERT | 98.74 | 90.60 |
| ELECTRA | 98.07 | 94.84 |
| DeBERTaV3 | 96.93 | 95.99 |

## B.5  OoD Hyperparameter Tuning

To avoid tuning on specific OoD datasets, we tune all detection methods using synthetic random noise as a validation set. ODIN's temperature scaling values are searched over $\{500.0, 1000.0, 2000.0\}$ and noise magnitudes in $\{0.0, 0.0005, 0.001, 0.0014, 0.002, 0.0024, 0.005, 0.01, 0.05, 0.1\}$ for both ODIN and Mahalanobis-based monitors.

## B.6  MIRA evaluation range selection

To make these perturbations comparable across models, we define the perturbation interval $[\epsilon_{\min}, \epsilon_{\max}]$ using a consistent strategy. Here, $\epsilon_{\min}$ is chosen as the smallest perturbation magnitude that reduces accuracy below a predefined threshold, and $\epsilon_{\max} = 2 \cdot \epsilon_{\min}$.

This construction allows to evaluate all models under a consistent notion of "boundary proximity". We use thresholds of $50\%$ for most datasets, $15\%$ for CIFAR-100, and $75\%$ for NLP tasks. Alterna-

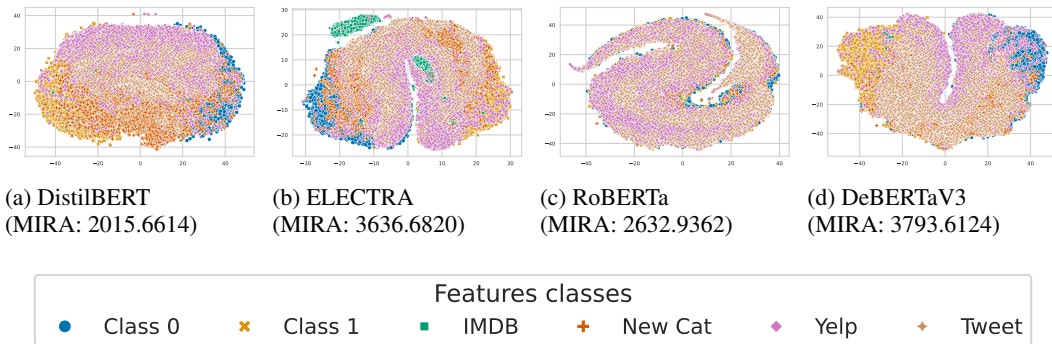

(a) DistilBERT
(MIRA: 2015.6614)

(b) ELECTRA
(MIRA: 3636.6820)

(c) RoBERTa
(MIRA: 2632.9362)

(d) DeBERTaV3
(MIRA: 3793.6124)

Features classes

● Class 0  ✕ Class 1  ■ IMDB  + New Cat  ◆ Yelp  ✦ Tweet

Figure 4: t-SNE visualization of penultimate-layer activations on NLP datasets. Cluster separation is harder to spot since all models do not have high OoD performance. However, by comparing the plots it is possible to see a better separation DeBERTaV3 and ELECTRA models with respect to RoBERTa and DistilBERT.

tive thresholds change absolute MIRA values but preserve model ordering, indicating that the metric is not overly sensitive to this choice. We note, however, that very high thresholds tend to produce less pronounced differences between models. Table 8 reports the resulting MIRA scores at different accuracy thresholds.

The key requirement is to apply the same threshold consistently across all models under comparison.

Finally, since no particular weighting of perturbation magnitudes is required, we adopt a uniform distribution $p(\epsilon) = \frac{1}{\epsilon_{\max} - \epsilon_{\min}}$, giving equal importance to all perturbations within the range.

## LLM USAGE

A large language model (ChatGPT, GPT-5 by OpenAI) was used in a limited way to refine text and improve readability. It did not contribute to the research design, experiments, or results.

Table 8: Evaluation range used to compute the MIRA score for all the models.

| Model | ID Data | Acc. threshold (%) | $\epsilon_{min}$ | MIRA |
|---|---|---|---|---|
| ResNet-18 | CIFAR10 | 65 | 0.0043 | 3.0076 |
| | | 50 | 0.012 | 6.0549 |
| | | 30 | 0.05 | 23.7248 |
| DenseNet | CIFAR10 | 65 | 0.0024 | 5.4914 |
| | | 50 | 0.004 | 16.0107 |
| | | 30 | 0.009 | 47.6899 |
| CustomNet | CIFAR10 | 65 | 0.0005 | −0.1277 |
| | | 50 | 0.002 | −0.07176 |
| | | 30 | 0.0075 | 0.4378 |
| ViT | CIFAR10 | 65 | 0.0018 | 49.5103 |
| | | 50 | 0.0038 | 89.2544 |
| | | 30 | 0.03 | 170.4944 |
| ResNet-18 | CIFAR100 | 50 | 0.00195 | 0.3569 |
| | | 30 | 0.005 | 0.6572 |
| | | 20 | 0.008 | 0.4894 |
| DenseNet | CIFAR100 | 50 | 0.00125 | 0.3688 |
| | | 30 | 0.0026 | 2.8060 |
| | | 20 | 0.0045 | 8.6561 |
| ViT | CIFAR100 | 50 | 0.001 | 16.1216 |
| | | 30 | 0.003 | 53.2344 |
| | | 20 | 0.008 | 80.8750 |
| MLP | Sensorless Drive Diagnosis | 65 | 0.195 | 25.1310 |
| | | 50 | 0.25 | 30.1594 |
| | | 30 | 0.36 | 50.3339 |
| DeepMLP | Sensorless Drive Diagnosis | 65 | 0.155 | 11.7452 |
| | | 50 | 0.2 | 12.7098 |
| | | 30 | 0.32 | 15.2353 |
| WideMLP | Sensorless Drive Diagnosis | 65 | 0.198 | 55.8145 |
| | | 50 | 0.26 | 63.5103 |
| | | 30 | 0.39 | 91.2932 |
| Transformer | Sensorless Drive Diagnosis | 65 | 0.28 | 5.3716 |
| | | 50 | 0.49 | 7.8706 |
| | | 30 | 0.97 | 8.3579 |
| DeepTransformer | Sensorless Drive Diagnosis | 65 | 0.22 | 3.4543 |
| | | 50 | 0.45 | 4.3676 |
| | | 30 | 0.92 | 7.4953 |
| RoBERTa | SST-2 | 85 | 0.0022 | 2296.7175 |
| | | 75 | 0.0038 | 2632.9362 |
| | | 65 | 0.0058 | 2832.9268 |
| ELECTRA | SST-2 | 85 | 0.0016 | 2242.7426 |
| | | 75 | 0.0032 | 3636.6820 |
| | | 65 | 0.005 | 4230.8988 |
| DistilBERT | SST-2 | 85 | 0.0004 | 634.1917 |
| | | 75 | 0.00085 | 2015.6614 |
| | | 65 | 0.0012 | 2734.1066 |
| DeBERTaV3 | SST-2 | 85 | 0.0028 | 2659.3444 |
| | | 75 | 0.005 | 3793.6124 |
| | | 65 | 0.0085 | 4494.2142 |

