# OpenReview forum: "MIRA: Quantifying Neural Network Monitorability via Feature Space Analysis"
_ICLR.cc/2026/Conference — Submitted to ICLR 2026_

### Official Review · Reviewer_BxyR · 2025-11-01

**Soundness:** 2
**Presentation:** 2
**Contribution:** 2
**Rating:** 4
**Confidence:** 3

**Summary:**

This work presents MIRA, a monitorability measure for neural networks based on feature space analysis. The proposed method works via input perturbations, and the Mahalanobis distance is used to measure the discrepancy between perturbed inputs and the training set in the feature space. Several experiments validate the proposed method.

**Strengths:**

The provided visualizations are clear and effectively demonstrate the alignment between the MIRA score and the "separability" of trained neural network features. This work also sheds light on the importance of monitorability in the deep neural network literature.

**Weaknesses:**

The connection between Section 3.2 and Section 3.3 is weak. Section 3.2 gives Definition 1 regarding abstract monitorability, while Section 3.3 provides an empirical metric serving as the main contribution of this paper. However, there is no guarantee nor clear theoretical connection establishing how the method given in Section 3.3 relates to Definition 1.

Furthermore, the data distribution $P_{in}$ in Definition 1 is somewhat ambiguous. Intuitively, what would be an example of $P_{in}$? If $P_{in}$ is the training distribution, then if the model achieves near-zero training loss, the $\ell$-monitorability becomes meaningless. On the other hand, if $P_{in}$ is supposed to be a mixture between training and (out-of-distribution) test distributions, how is this distribution reflected by only the training set (potentially with perturbation)?

**Questions:**

Please see the weaknesses above. Minor questions:

- The formulation in line 175 is very similar to some kind of adversarial robustness. Are there any links between Equation (2) and other adversarial robustness metrics?
- What is $\tilde{x}$ in line 187?

---

> ### Author Response · Authors · 2025-11-12
>
> We thank the reviewer for their constructive comments and appreciation of the relevance of our contribution. We address the main concerns below.
>
> ### 1. Connection between Definition 1 and MIRA score
> We agree that the link could be more explicitly stated. Definition 1 (monitorability) is a theoretical, binary notion.
> Section 3.3 (MIRA Score) is an operational, quantitative measure that serves as a proxy for monitorability by quantifying how perturbations affect the feature-space distances between perturbed and unperturbed inputs. In a similar way, correctness (theoretical, binary) of a model is evaluated by measuring its accuracy (operational, quantitative).
> In this sense, the empirical MIRA score instantiates Definition 1 using measurable quantities (Mahalanobis distances and perturbation magnitude). We will clarify this conceptual alignment in the revised version by explicitly referencing Definition 1 when introducing the metric, and by adding a bridging paragraph at the start of Section 3.3.
>
> ### 2. Ambiguity of the distribution $P_{in}$
> We appreciate the opportunity to clarify this. $P_{in}$ denotes the ID data distribution, which is the one the model is trained on.
> The loss term in Definition 1 is used to indicate "correct inference", not as a training objective. In essence, Definition 1 states that if the loss is low enough (i.e., correct inference) then the internal representation is in $Z^l$, and vice versa.
> If the training loss is zero, this means we have correct inference but does not alter the definition.
> On anomalous data or misclassified inputs the loss is inevitably high, and the definition requires the resulting features to lie outside $Z^l$.
>
> ### 3. Relation to adversarial robustness
> We thank the reviewer for the question. Yes, there is a conceptual connection: both MIRA and adversarial robustness consider the model’s response to input perturbations. However, MIRA is diagnostic, not defensive. It does not aim to find worst-case adversarial examples or evaluate robustness margins, but rather measures how features respond to perturbations. Thus, a network can be non-robust yet still highly monitorable.
>
> Finally, $\tilde{x}$ in line 187 should simply be $x$ (generic input to the network). We will fix this in a revision.

---

> ### Author Response · Authors · 2025-11-19
>
> ### Additional remark on the link between Definition 1 and the empirical metric
> We would like to emphasize that the relationship between our formal definition of monitorability and the MIRA score mirrors existent patterns also in the robustness literature [2]. For example, in adversarial robustness, formal definitions (e.g., robustness under norm-bounded perturbations as in [1] or [3]) provide high-level properties of interest, while the practical metrics used to approximate or evaluate robustness (e.g., margins, local Lipschitz estimates, empirical worst-case perturbations) can vary in form and implementation. Despite these differences, the metrics still aim to operationalize the same underlying notion of robustness.
>
> In an analogous way, Definition 1 specifies monitorability as an abstract property of a neural network, while the MIRA score provides a concrete and computable instantiation of that property. The metric is not meant to mirror the definition in form, but to capture the core idea: the stability and separability of internal representations under perturbations.
>
> We thank the reviewer again for their valuable feedback and hope this clarification sheds further light on the motivation behind our choice.
>
> [1] Evaluating the Robustness of Neural Networks: An Extreme Value Theory Approach
>
> [2] On the Robustness of Explanations of Deep Neural Network Models: A Survey
>
> [3] Robustness assessment and improvement of a neural network for blood oxygen pressure estimation

---

> ### Author Response · Authors · 2025-11-20
>
> For completeness, we include a proof sketch below to support our claims and provide further technical clarity for the reviewers.
>
> ### Soundness of the MIRA Score
>
> While Definition 1 formalizes monitorability as a binary property,
> the MIRA score provides a graded, empirical quantification of how strongly this
> property holds in practice. In this section, we establish a formal
> correspondence between the two notions, showing that high MIRA values arise
> precisely when the condition of Definition 1 is satisfied with a
> sufficient margin. Conversely, low MIRA values indicate a violation of this
> margin and therefore low monitorability.
>
> **Setup.**
>
> Let $h : X \to R^{n_l}$ denote the representation map at layer $l$, i.e.,
> $h(x) = f^l(x)$.
> Let $Z^l \subseteq R^{n_l}$ be the set from Definition 1
> representing the ``trusted'' region of feature space associated with
> low-loss predictions.
> We assume that the ID feature distribution for each class is approximately
> Gaussian, as commonly done in Mahalanobis-based uncertainty estimation [1].
> Let $D$ denote an empirical dataset drawn from $\mathcal{P}_{in}$.
>
> **Monitorability margin.**
>
> Definition 1 requires that correct (low-loss) predictions
> correspond to representations in $Z^l$.
> To quantify the strength of this property, we define the *monitorability
> margin*:
> \begin{equation}
>     \gamma =
>     \inf_{x : \mathcal{L}(f(x),y)\le\epsilon_{loss}}
>     \mathrm{dist}_M\big(h(x), Z^{l\,c}\big),
> \end{equation}
> where $\mathrm{dist}_M$ denotes Mahalanobis distance with respect to the
> class-conditional covariance.
> Intuitively, $\gamma$ measures how strongly the representations of correctly
> classified inputs are separated from those corresponding to erroneous
> predictions.
>
> **Perturbation model.**
> Recall that MIRA evaluates how far controlled perturbations
> $x \mapsto x+\delta(x,\epsilon)$ push representations away from the ID region.
> We assume, as in standard adversarial analyses, that:
> (i) $h$ is $L$-Lipschitz, and
> (ii) $\delta(x,\epsilon)$ moves $x$ toward the decision boundary of $f$.
>
> **Theorem 1** *(Soundness of MIRA as a Monitorability Estimator)*
>     Let $f$ be $l$-monitorable with margin $\gamma>0$ in the sense above.
>     Assume that $h$ is $L$-Lipschitz and that perturbed inputs
>     $x+\delta(x,\epsilon)$ eventually cross the decision boundary at some magnitude
>     $\epsilon^* \in [\epsilon_{\min}, \epsilon_{\max}]$.
>     Then there exist constants $c, c' > 0$ such that:
>     \begin{equation}
>         S\big(h(x+\delta(x,\epsilon^*))\big) - S_0
>         \ge c\,\gamma
>         \qquad\text{and}\qquad
>         MIRA(f,D,l) \ge c'\,\gamma.
>     \end{equation}
>
> *Proof sketch*
> If $f$ is $l$-monitorable with margin $\gamma$, then all correct predictions
> satisfy $h(x) \in Z^l$ and their distance to $Z^{l\,c}$ is at least $\gamma$.
> When a perturbation crosses the decision boundary, we have
> $h(x+\delta(x,\epsilon^*)) \in Z^{l\,c}$.
> By Lipschitzness of $h$ and the Gaussian feature model, leaving $Z^l$ by at
> least $\gamma$ increases the Mahalanobis-based surprisal by an amount
> proportional to $\gamma$.
> Since MIRA integrates the expected surprisal increase over perturbations, the
> lower bound on the pointwise surprisal translates directly into the stated
> lower bound on $MIRA(f,D,l)$.
>
> **Interpretation.**
>
> Theorem 1 shows that when the representation space separates correct and erroneous predictions (as required by
> Definition 1), perturbed inputs produce feature vectors that
> move decisively away from the ID region, resulting in a *large* MIRA
> score.
> Conversely, if MIRA is small, then perturbed features remain close to the clean
> ID region, implying that the boundary between correct and erroneous
> representations is weak and that monitorability is violated.
> Thus, MIRA serves as a principled empirical estimator of the monitorability
> property formalized in Definition 1.
>
>
> [1] A Simple Unified Framework for Detecting Out-of-Distribution Samples and Adversarial Attacks

---

> ### Author Response · Authors · 2025-11-27
>
> Dear Reviewer BxyR,
>
> We hope this message finds you well. Thank you again for the thoughtful and constructive comments you provided in your review. As the discussion deadline on December 3 is approaching, we would greatly appreciate it if you could take a moment to check whether our rebuttal has resolved your points. Your insights are valuable to us, and if anything remains unclear or if further clarification would be helpful, we would be more than happy to provide it.
>
> Thank you sincerely for your time and effort in reviewing our work.
>
> Best regards, The Authors

---

### Official Review · Reviewer_w33X · 2025-11-01

**Soundness:** 1
**Presentation:** 3
**Contribution:** 1
**Rating:** 2
**Confidence:** 5

**Summary:**

The paper proposes the concept of monitorability, whereby the inference maybe assigned a score that showcases whether said inference maybe trusted. The authors claim that existing uncertainty estimates are based on well-trained networks and features. Additionally, OOD estimation generally separates ID from OOD without addressing the correctness of ID sample inference. The authors propose MIRA, a method to monitor neural network predictions.

**Strengths:**

1. The paper is well written with a clear logical flow.

2. The results are showcased on a variety of datasets.

**Weaknesses:**

**Incorrect motivation and claims**:

The paper makes some fundamentally incorrect claims:

1. (Line 12) *Although out-of-distribution (OOD) detection and uncertainty estimation have been widely studied, they often rely on the assumption that neural networks learn high-quality features.*: This is false. Early-stopping is a well known method in UQ that explicitly states that network training must stop long before overfitting.

2. (Line 87) *However, it is important to note that misclassifications may also occur for ID inputs, which is a distinct scenario not directly addressed by OoD detection.*: True. But the application that does look at ID classification and OOD detection is Open set recognition [1].

3. (Line 57) *To the best of our knowledge, this is the first work to formally define and quantify monitorability as a distinct property of neural networks.*: True in the sense of using the word monitorability. But there are a large number of simple UQ metrics (margin sampling, entropy), and more complex internal state-based gradient metrics [2, 3] that can monitor the outputs and provide an alternative score.

[1] Recent Advances in Open Set Recognition: A Survey

[2] Probing the Purview of Neural Networks via Gradient Analysis

[3] Counterfactual Gradients-based Quantification of Prediction Trust in Neural Networks

**Paper positioning**: Without referencing and comparing against prediction trust, UQ, open set recognition, it is hard to evaluate the contributions and results in the paper.

**Questions:**

Please note that my rating is based on the weaknesses above. I would be willing to reevaluate my rating if I have missed something fundamental about the concept of monitoriability that is not covered by UQ, prediction trust, or open set recognition.

---

> ### Author Response · Authors · 2025-11-12
>
> We thank the reviewer for their feedback and the opportunity to clarify our motivation and contribution.
>
> ### 1. High-quality features assumption
> Early stopping is indeed a mitigation technique for overfitting but it does not guarantee feature robustness or monitorability. Our statement refers to the assumption underlying OOD and UQ methods, not to training heuristics.
> OOD/UQ methods assume that the trained network’s latent space cleanly separates classes and that internal features are semantically meaningful. For example, using Mahalanobis distance explicitly relies on separable internal features to distinguish ID from OOD/anomalous inputs.
>
> ### 2. ID misclassifications and Open Set Recognition
> OSR deals with detecting unknown classes at test time, i.e., distinguishing “known” vs. “unknown” classes. Misclassifications within the known (ID) set, such as confusing two known classes, are not the same as open-set errors.
>
> ### 3. Relation to uncertainty and trust metrics.
> We appreciate the reviewer’s references. These works introduce per-sample trust scores, that can eventually be thresholded and used for detection. However, they are not concerned with how amenable a model is to monitoring across inputs. Our contribution lies precisely here: we propose a formal definition and empirical quantification of monitorability as a distinct, intrinsic property of a neural network. Importantly, our score is computed over the ID data, not per sample. It is not a detection score and cannot be used to separate samples, but rather an aggregated measure. We will expand the related work and discuss these distinctions explicitly in the revision.
>
> We hope this clarifies the novelty and scope of our contribution relative to existing UQ and trust methods.

---

> ### Author Response · Authors · 2025-11-19
>
> ### Remark on the positioning of our work
> Our goal is not to perform detection or uncertainty estimation directly, but to assess whether such methods can, in principle, be effective on a given neural network. MIRA evaluates the intrinsic ability of the model to keep a well disentangled internal representation that supports reliable monitoring. In other words, rather than proposing a new monitoring or uncertainty estimation technique, we quantify the potential amenability of a trained model to these downstream approaches. This positions our work as complementary and foundational to detection and uncertainty estimation research, providing a means to evaluate the conditions under which such methods can succeed.
>
> We thank the reviewer again and remain available for any follow-up questions should any aspect of our clarification be incomplete.

---

> ### Author Response · Authors · 2025-11-20
>
> For completeness, we include a proof sketch below to support our claims and provide further technical clarity for the reviewers.
>
> ### Soundness of the MIRA Score
>
> While Definition 1 formalizes monitorability as a binary property,
> the MIRA score provides a graded, empirical quantification of how strongly this
> property holds in practice. In this section, we establish a formal
> correspondence between the two notions, showing that high MIRA values arise
> precisely when the condition of Definition 1 is satisfied with a
> sufficient margin. Conversely, low MIRA values indicate a violation of this
> margin and therefore low monitorability.
>
> **Setup.**
>
> Let $h : X \to R^{n_l}$ denote the representation map at layer $l$, i.e.,
> $h(x) = f^l(x)$.
> Let $Z^l \subseteq R^{n_l}$ be the set from Definition 1
> representing the ``trusted'' region of feature space associated with
> low-loss predictions.
> We assume that the ID feature distribution for each class is approximately
> Gaussian, as commonly done in Mahalanobis-based uncertainty estimation [1].
> Let $D$ denote an empirical dataset drawn from $\mathcal{P}_{in}$.
>
> **Monitorability margin.**
>
> Definition 1 requires that correct (low-loss) predictions
> correspond to representations in $Z^l$.
> To quantify the strength of this property, we define the *monitorability
> margin*:
> \begin{equation}
>     \gamma =
>     \inf_{x : \mathcal{L}(f(x),y)\le\epsilon_{loss}}
>     \mathrm{dist}_M\big(h(x), Z^{l\,c}\big),
> \end{equation}
> where $\mathrm{dist}_M$ denotes Mahalanobis distance with respect to the
> class-conditional covariance.
> Intuitively, $\gamma$ measures how strongly the representations of correctly
> classified inputs are separated from those corresponding to erroneous
> predictions.
>
> **Perturbation model.**
> Recall that MIRA evaluates how far controlled perturbations
> $x \mapsto x+\delta(x,\epsilon)$ push representations away from the ID region.
> We assume, as in standard adversarial analyses, that:
> (i) $h$ is $L$-Lipschitz, and
> (ii) $\delta(x,\epsilon)$ moves $x$ toward the decision boundary of $f$.
>
> **Theorem 1** *(Soundness of MIRA as a Monitorability Estimator)*
>     Let $f$ be $l$-monitorable with margin $\gamma>0$ in the sense above.
>     Assume that $h$ is $L$-Lipschitz and that perturbed inputs
>     $x+\delta(x,\epsilon)$ eventually cross the decision boundary at some magnitude
>     $\epsilon^* \in [\epsilon_{\min}, \epsilon_{\max}]$.
>     Then there exist constants $c, c' > 0$ such that:
>     \begin{equation}
>         S\big(h(x+\delta(x,\epsilon^*))\big) - S_0
>         \ge c\,\gamma
>         \qquad\text{and}\qquad
>         MIRA(f,D,l) \ge c'\,\gamma.
>     \end{equation}
>
> *Proof sketch*
> If $f$ is $l$-monitorable with margin $\gamma$, then all correct predictions
> satisfy $h(x) \in Z^l$ and their distance to $Z^{l\,c}$ is at least $\gamma$.
> When a perturbation crosses the decision boundary, we have
> $h(x+\delta(x,\epsilon^*)) \in Z^{l\,c}$.
> By Lipschitzness of $h$ and the Gaussian feature model, leaving $Z^l$ by at
> least $\gamma$ increases the Mahalanobis-based surprisal by an amount
> proportional to $\gamma$.
> Since MIRA integrates the expected surprisal increase over perturbations, the
> lower bound on the pointwise surprisal translates directly into the stated
> lower bound on $MIRA(f,D,l)$.
>
> **Interpretation.**
>
> Theorem 1 shows that when the representation space separates correct and erroneous predictions (as required by
> Definition 1), perturbed inputs produce feature vectors that
> move decisively away from the ID region, resulting in a *large* MIRA
> score.
> Conversely, if MIRA is small, then perturbed features remain close to the clean
> ID region, implying that the boundary between correct and erroneous
> representations is weak and that monitorability is violated.
> Thus, MIRA serves as a principled empirical estimator of the monitorability
> property formalized in Definition 1.
>
>
> [1] A Simple Unified Framework for Detecting Out-of-Distribution Samples and Adversarial Attacks

---

> ### Author Response · Authors · 2025-11-26
>
> We would also like to clarify and strengthen our motivation regarding the implicit assumption of high-quality features. Several recent works explicitly highlight that the effectiveness of OoD and anomaly detection critically depends on the quality and structure of the learned feature space. For example, [1] show that representation misalignment and feature-level distortions substantially degrade OOD detection performance, indicating that many detectors presuppose well-separated and semantically meaningful latent features. Similarly, [2] develop a unified clustering-based anomaly detection framework and demonstrate that anomaly separability hinges on the consistency of internal embeddings. Furthermore, [3] finds that enforcing neural collapse (i.e., highly structured and well-separated class features) directly improves OoD detection, illustrating that the success of such methods relies on the latent space exhibiting strong geometric structure.
>
> These works collectively reinforce our point: although monitoring methods are widely studied, they their effectiveness depends on neural networks learning high-quality features.
>
> [1] Neural Collapse Inspired Feature Alignment for Out-of-Distribution Generalization
> [2] Towards a Unified Framework of Clustering-based Anomaly Detection
> [3] Controlling Neural Collapse Enhances Out-of-Distribution Detection and Transfer Learning

---

> ### Author Response · Authors · 2025-11-27
>
> Dear Reviewer w33X,
>
> We hope this message finds you well. Thank you again for the thoughtful and constructive comments you provided in your review. As the discussion deadline on December 3 is approaching, we would greatly appreciate it if you could take a moment to check whether our rebuttal has resolved your points. Your insights are valuable to us, and if anything remains unclear or if further clarification would be helpful, we would be more than happy to provide it.
>
> Thank you sincerely for your time and effort in reviewing our work.
>
> Best regards,
> The Authors

---

### Official Review · Reviewer_SrJp · 2025-11-09

**Soundness:** 2
**Presentation:** 3
**Contribution:** 3
**Rating:** 6
**Confidence:** 4

**Summary:**

This paper attempts to relax the implicit assumption that most black-box OOD and UQ approaches make, which is that models have learned semantically meaningful features. In practice, if this assumption does not hold, it may lead to undetected errors. The authors formalize the concept of monitorability and an associated MIRA score, which is intended to highlight inference errors at the internal model layer level, rather than merely relying on a black-box assessment. The score quantifies this property by applying norm-bounded input perturbations and measuring the separability of resulting feature representations using Mahalanobis distance. MIRA is validated across computer vision, tabular, and NLP tasks by demonstrating correlation with OOD detection performance.

The paper addresses an important problem for safety-critical applications and makes a valuable theoretical contribution, however, the work has limitations: (1) MIRA's layer-dependency lacks principled aggregation to provide an overall monitorability estimate, the perturbation range selection requires domain-specific threshold choices, and the validation has some circularity.

**Strengths:**

This paper has the following strengths:

Formalizing monitorability as a distinct property of networks is valuable and contributes to trustworthiness and explanability that is sought from today's models (I think integrating formal methods and verification, as pointed out as extensions, is an interesting direction too).

The use of Mahalanobis distance with the surprisal score normalization (Eq 3) appropriately handles dimensionality differences, which would have been a concern if not addressed.

Experiments span three data modalities: vision, tabular data, and NLP with diverse architectures, demonstrating consistent correlation with OOD detection performance.

MIRA captures monitoring of internal model semantics which has a  potential to be applied independently of any specific detection method.

**Weaknesses:**

There are two primary weaknesses of the approach:

The primary weakness of the proposed score is that it is layer l dependent and there is a lack of aggregation of this score at the layer level to provide an overall aggregate estimate of how the effect of cascading layers affects overall monitorability of the network as a whole. Specifically, the missing multi-layer analysis has huge architectural dependence and implications. The experimental results show comparisons for the monitoring methods across different classes, but not across different layers. The authors should: (1) provide empirical evidence and theoretical justification for selecting a specific layer (e.g., final layer) as representative or (2) develop a principled aggregation scheme that accounts for the cascading effects of these networks. Investigating how MIRA scores vary across different layers within the same network would be a valuable insight.

The claim of not requiring external OOD is a lucrative one, however, it is undermined by the requirement for a user-defined distribution for perturbation magnitudes p(epsilon) in definition 2 on page 4. While Appendix B.6 proposes a threshold-based heuristic (selecting epsilon to reduce accuracy below pre-determined thresholds), this approach introduces domain-dependent choices (50% for some datasets, 15% for CIFAR-100, and 75% for NLP) without principled guidance on how this should be selected in practice. Since perturbation magnitudes may vary significantly across test domains, the lack of a principled selection strategy for p(epsilon) represents a critical gap. The authors should provide: (1) a theoretical or empirical justification for p(epsilon) choices and/or (2) an adaptive method for automatically determining p(epsilon) across datasets.

**Questions:**

On page 2, the alignment of the Mahalanobis distance with the softmax classifier remains specific to this type of classifier. Does this assumption hold for other types of classifiers? The broader question is about the generalization of the method to other model architectures.

On page 4, the perturbation delta(x,epsilon) needs to move x toward the decision boundary -> how is this perturbation selected in practice? Taking this further, in definition 2 on the same page, p(epsilon) is a user-defined probability distribution over perturbation magnitudes. It is not clear as to how the user should select this probability distribution. Are there any insights here that can be leveraged?

In practice, the magnitude of perturbations is not known and hence this is a critical assumption. So, while no external OOD data is required, the perturbation magnitude distribution, which is assumed to be user-specified, is unknown and a principled method to obtain this should be explored further and formalized. While section 4.2 on page 5 provides details on the use of FGSM to obtain perturbations, it is not clear as to why the authors "consider this as a better choice."

The authors state on page 4 that MIRA is not intended for runtime detection, but rather as a pre-deployment evaluation metric. It is not clear, then, how this is used downstream at runtime. Can models be explicitly trained to improve their MIRA scores?

The evaluation creates a somewhat circular argument: the MIRA metric is validated by showing it correlates with detection performance, but monitorability is itself defined as detectability. I understand there is a lack of baselines in the space, however, is there a theoretical justification that can be provided here to back up the results? Addressing the multi-layer analysis would help.

No error bars or confidence intervals are provided for MIRA scores or OOD detection performance. Given the sensitivity shown in Table 8, understanding variance is important.

---

> ### Author Response · Authors · 2025-11-12
>
> We thank the reviewer for the detailed and constructive feedback. We address the main points below.
>
> ### 1. Layer dependency and aggregation
> We agree that the layer-wise nature of MIRA deserves further discussion. Our formulation is intentionally layer-specific, as monitorability is defined with respect to the semantic coherence of internal representations, which differ across layers. Aggregating scores across heterogeneous layers would conflate representations serving distinct roles (e.g., low-level vs. high-level features). Furthermore, most monitoring techniques rely on single-layer monitoring (usually the last hidden layer): in this context, we also note that MIRA itself could be used as a metric to choose which layer to monitor. We will expand on this in a revision.
> Nonetheless, we acknowledge that understanding how monitorability evolves across layers is valuable and we mention it as potential evolution of this work.
>
> ### 2. Choice of $p(\epsilon)$
> We thank the reviewer for this observation and agree that our notation may have suggested more freedom than intended. In practice, $p(\epsilon)$ is just a weighting function over the perturbation range [$\epsilon_{min}, \epsilon_{max}$]. The relevant parameter here is $\epsilon_{min}$, which determines the lower bound for the perturbation magnitude at which performance begins to degrade. The choice of this bound is indeed domain-dependent. However, we show in Appendix B.6 that different values for $\epsilon_{min}$ still lead to consistent results, as long as the threshold is the same. A general guideline is to decrease the accuracy threshold as the number of classes increases. We will be happy to clarify it in the revised version.
>
> ### 3. Applicability beyond softmax classifiers
> MIRA operates on internal features and only requires class-conditional means and covariances. The use of Mahalanobis distance is independent of the classifier type and extends naturally to architectures without a softmax output layer.
>
> ### 4. Perturbation method (FGSM)
> FGSM is chosen for its efficiency and directionality, since it produces controlled perturbations toward the decision boundary, aligning with the definition’s goal of probing local sensitivity rather than adversarial success. Therefore, the $\delta(x,\epsilon)$ in our definition is generated with FGSM in practice. However, in principle other perturbations can also be used (e.g. PDG or C&W) without affecting the meaning of the metric.
>
> ### 5. Runtime applicability
> We clarify that MIRA is intended as a pre-deployment diagnostic metric to assess a model’s inherent monitorability, not as a runtime monitor itself. Models can be compared or fine-tuned to improve their MIRA scores before deployment.
>
> ### 6. Correlation and evaluation
> We acknowledge that monitorability and detectability are related concepts. However, our validation is not circular: monitorability depends on the network's behaviour at a certain layer. Detection performance also depends on the specific monitoring method.
> In our experiments, we take the best performance across three different monitoring methods, and use this as a proxy for an "ideal monitor".
> The correlation of this proxy with MIRA supports that the latter captures the model's ability to generate a well-structed internal representation that is suitable for monitoring.
>
> ### 7. Variance and confidence intervals
> We appreciate the reviewer’s attention to robustness. In our setup, there are no stochastic sources affecting the MIRA computation: FGSM perturbations are fully deterministic given the model parameters, and the $\epsilon$ values in the integral are fixed.
> The weighting function $p(\epsilon)$ simply scales contributions across $\epsilon$ and does not introduce randomness. Consequently, both the MIRA score and the reported OOD metrics are deterministic for a given trained model. We will clarify this in the revised version and adjust the notation of $p(\epsilon)$ to emphasize its role as a weighting function, rather than a probabilistic distribution.
>
> We hope this helps with addressing the reviewer's concerns and clarify our contributions.

---

> ### Author Response · Authors · 2025-11-20
>
> For completeness, we include a proof sketch below to support our claims and provide further technical clarity for the reviewers.
>
> ### Soundness of the MIRA Score
>
> While Definition 1 formalizes monitorability as a binary property,
> the MIRA score provides a graded, empirical quantification of how strongly this
> property holds in practice. In this section, we establish a formal
> correspondence between the two notions, showing that high MIRA values arise
> precisely when the condition of Definition 1 is satisfied with a
> sufficient margin. Conversely, low MIRA values indicate a violation of this
> margin and therefore low monitorability.
>
> **Setup.**
>
> Let $h : X \to R^{n_l}$ denote the representation map at layer $l$, i.e.,
> $h(x) = f^l(x)$.
> Let $Z^l \subseteq R^{n_l}$ be the set from Definition 1
> representing the ``trusted'' region of feature space associated with
> low-loss predictions.
> We assume that the ID feature distribution for each class is approximately
> Gaussian, as commonly done in Mahalanobis-based uncertainty estimation [1].
> Let $D$ denote an empirical dataset drawn from $\mathcal{P}_{in}$.
>
> **Monitorability margin.**
>
> Definition 1 requires that correct (low-loss) predictions
> correspond to representations in $Z^l$.
> To quantify the strength of this property, we define the *monitorability
> margin*:
> \begin{equation}
>     \gamma =
>     \inf_{x : \mathcal{L}(f(x),y)\le\epsilon_{loss}}
>     \mathrm{dist}_M\big(h(x), Z^{l\,c}\big),
> \end{equation}
> where $\mathrm{dist}_M$ denotes Mahalanobis distance with respect to the
> class-conditional covariance.
> Intuitively, $\gamma$ measures how strongly the representations of correctly
> classified inputs are separated from those corresponding to erroneous
> predictions.
>
> **Perturbation model.**
> Recall that MIRA evaluates how far controlled perturbations
> $x \mapsto x+\delta(x,\epsilon)$ push representations away from the ID region.
> We assume, as in standard adversarial analyses, that:
> (i) $h$ is $L$-Lipschitz, and
> (ii) $\delta(x,\epsilon)$ moves $x$ toward the decision boundary of $f$.
>
> **Theorem 1** *(Soundness of MIRA as a Monitorability Estimator)*
>     Let $f$ be $l$-monitorable with margin $\gamma>0$ in the sense above.
>     Assume that $h$ is $L$-Lipschitz and that perturbed inputs
>     $x+\delta(x,\epsilon)$ eventually cross the decision boundary at some magnitude
>     $\epsilon^* \in [\epsilon_{\min}, \epsilon_{\max}]$.
>     Then there exist constants $c, c' > 0$ such that:
>     \begin{equation}
>         S\big(h(x+\delta(x,\epsilon^*))\big) - S_0
>         \ge c\,\gamma
>         \qquad\text{and}\qquad
>         MIRA(f,D,l) \ge c'\,\gamma.
>     \end{equation}
>
> *Proof sketch*
> If $f$ is $l$-monitorable with margin $\gamma$, then all correct predictions
> satisfy $h(x) \in Z^l$ and their distance to $Z^{l\,c}$ is at least $\gamma$.
> When a perturbation crosses the decision boundary, we have
> $h(x+\delta(x,\epsilon^*)) \in Z^{l\,c}$.
> By Lipschitzness of $h$ and the Gaussian feature model, leaving $Z^l$ by at
> least $\gamma$ increases the Mahalanobis-based surprisal by an amount
> proportional to $\gamma$.
> Since MIRA integrates the expected surprisal increase over perturbations, the
> lower bound on the pointwise surprisal translates directly into the stated
> lower bound on $MIRA(f,D,l)$.
>
> **Interpretation.**
>
> Theorem 1 shows that when the representation space separates correct and erroneous predictions (as required by
> Definition 1), perturbed inputs produce feature vectors that
> move decisively away from the ID region, resulting in a *large* MIRA
> score.
> Conversely, if MIRA is small, then perturbed features remain close to the clean
> ID region, implying that the boundary between correct and erroneous
> representations is weak and that monitorability is violated.
> Thus, MIRA serves as a principled empirical estimator of the monitorability
> property formalized in Definition 1.
>
>
> [1] A Simple Unified Framework for Detecting Out-of-Distribution Samples and Adversarial Attacks

---

> ### Author Response · Authors · 2025-11-27
>
> Dear Reviewer SrJp,
>
> We hope this message finds you well. Thank you again for the thoughtful and constructive comments you provided in your review. As the discussion deadline on December 3 is approaching, we would greatly appreciate it if you could take a moment to check whether our rebuttal has resolved your points. Your insights are valuable to us, and if anything remains unclear or if further clarification would be helpful, we would be more than happy to provide it.
>
> Thank you sincerely for your time and effort in reviewing our work.
>
> Best regards,
> The Authors

---

### Meta-Review · Area_Chair_cYAF · 2025-12-18

**Summary:**

This paper was reviewed by three experts in the field, and the reviews are mixed. The paper received scores of Marginal Accept (6), Reject (2), and Marginal Reject (4).

This article discusses the limitations of existing methods for quantifying uncertainty and detecting OOD, which often implicitly assume that neural networks have learned semantically meaningful internal representations. When this assumption fails, these approaches may be unable to detect erroneous predictions, which poses risks in safety-critical applications. To address this problem, the authors introduce the concept of “monitorability,” which characterizes a model's intrinsic ability to reveal potential inference errors through its internal activations rather than solely through “black box” outputs.
The paper formalizes monitorability and proposes the MIRA (Monitorability via Input Perturbation) score, a practical metric that quantifies this property without requiring access to external OOD data. The MIRA score works by applying norm-bounded input perturbations and measuring the separability of the resulting feature representations in the model's latent space using Mahalanobis distance. The method is applicable to different layers and modalities.

Based on the reviews, I side with the reviewers recommending rejection. The authors are encouraged to carefully consider the reviewers’ comments, to improve the paper for submission elsewhere.

**Reviewer Concerns:**

The reviewers raised several important concerns that can be summarized as follows:
Conceptual Positioning and Relation to Prior Work:
 1. Although the article claims to introduce a new field called “monitorability,” the reviewers questioned whether this concept was truly distinct from existing notions of uncertainty quantification. I would add that it is really difficult to see whether the author actually answers to RQ1. For example, only 4 DNNs and 3 OOD criteria are used, with no distinction between near OOD and far OOD. Often, techniques that work well for near OOD do not work well for far OOD. It would have been interesting to have a proper and clear evaluation with more OOD detection strategies such as SCALE, ASH, React, VIM, MCP, Max Logit, KNN, etc. The authors also claim that they can help quantify uncertainty, but no experiments have been conducted with regard to random uncertainty. There are also a lot of works like for example [1] that focus on neural collapse and latent space and relate it to OOD.
2. Theoretical Gaps and Definition–Metric Disconnect:
Multiple reviewers highlighted a really weak connection between the abstract definition of monitorability (Definition 1) and the proposed empirical metric (MIRA).
3. Layer Dependence and Missing Aggregation:
The MIRA score is layer-dependent, yet the paper does not provide a principled way to aggregate monitorability across layers or justify why a specific layer should be representative of the network as a whole.
4. Evaluation Concerns and Circular Validation:
Although the experiments cover several modalities and show correlations between MIRA and OOD detection performance, the reviewers noted the absence of uncertainty estimates. I would also like to add that aleatoric uncertainty quantification is not evaluated, which makes it difficult to assess the various claims made in the article.

[1] Ammar, Mouïn Ben, et al. "Double Descent Meets Out-of-Distribution Detection: Theoretical Insights and Empirical Analysis on the role of model complexity." arXiv preprint arXiv:2411.02184 (2024).

**Reviewer Scores:**

The initial ratings were  Marginal Accept (6), Reject (2), and Marginal Reject (4). I think that after the rebuttal, the authors responded to some points, but not all. They also state that they will improve the revised version without correcting the current version in openreview. The authors have therefore not addressed all the points. After reading the article, I believe it cannot be accepted as it stands.

---

### Decision · Program_Chairs · 2026-01-26

Reject